# Microplastics and Kidneys: An Update on the Evidence for Deposition of Plastic Microparticles in Human Organs, Tissues and Fluids and Renal Toxicity Concern

**DOI:** 10.3390/ijms241814391

**Published:** 2023-09-21

**Authors:** Edoardo La Porta, Ottavia Exacoustos, Francesca Lugani, Andrea Angeletti, Decimo Silvio Chiarenza, Carolina Bigatti, Sonia Spinelli, Xhuliana Kajana, Andrea Garbarino, Maurizio Bruschi, Giovanni Candiano, Gianluca Caridi, Nicoletta Mancianti, Marta Calatroni, Daniela Verzola, Pasquale Esposito, Francesca Viazzi, Enrico Verrina, Gian Marco Ghiggeri

**Affiliations:** 1UO Nephrology Dialysis and Transplant, IRCCS Istituto Giannina Gaslini, 16147 Genoa, Italy; edoardo01laporta@gmail.com (E.L.P.); ottaviaexacoustos1994@gmail.com (O.E.); francescalugano@gaslini.com (F.L.); andreaangeletti@gaslini.org (A.A.); silvio.chiarenza@yahoo.com (D.S.C.); carolina.bigatti.cb@gmail.com (C.B.); enricoverrina@gaslini.org (E.V.); 2UOSD Dialysis IRCCS Istituto Giannina Gaslini, 16147 Genoa, Italy; giovannicandiano@gaslini.org (G.C.); gianlucacaridi@gaslini.org (G.C.); 3Laboratory of Molecular Nephrology, IRCCS Istituto Giannina Gaslini, 16147 Genoa, Italy; soniaspinelli@gaslini.org (S.S.); xhuliakajana@gmail.com (X.K.); andreagarbarino@gmail.com (A.G.); mauriziobruschi@yahoo.it (M.B.); 4Department of Experimental Medicine (DIMES), University of Genoa, 16132 Genoa, Italy; 5Nephrology, Dialysis and Transplantation Unit, Department of Emergency-Urgency and Transplantation, University Hospital of Siena, 53100 Siena, Italy; mancianti25121988@gmail.com; 6Department of Biomedical Sciences, Humanitas University, Pieve Emanuele, 20072 Milan, Italy; marta.calatroni@hunimed.eu; 7Nephrology and Dialysis Division, IRCCS Humanitas Research Hospital, Rozzano, 20089 Milan, Italy; 8Department of Internal Medicine, University of Genoa, 16132 Genoa, Italy; daverz@libero.it (D.V.); pasquale.esposito@unige.it (P.E.); francesca.viazzi@unige.it (F.V.); 9Division of Nephrology, Dialysis and Transplantation IRCCS Ospedale Policlinico San Martino, 16132 Genoa, Italy

**Keywords:** microplastics, environment, toxicity, kidney disease, Raman spectroscopy

## Abstract

Plastic pollution became a main challenge for human beings as demonstrated by the increasing dispersion of plastic waste into the environment. Microplastics (MPs) have become ubiquitous and humans are exposed daily to inhalation or ingestion of plastic microparticles. Recent studies performed using mainly spectroscopy or spectrometry-based techniques have shown astounding evidence for the presence of MPs in human tissues, organs and fluids. The placenta, meconium, breast milk, lung, intestine, liver, heart and cardiovascular system, blood, urine and cerebrovascular liquid are afflicted by MPs’ presence and deposition. On the whole, obtained data underline a great heterogeneity among different tissue and organs of the polymers characterized and the microparticles’ dimension, even if most of them seem to be below 50–100 µm. Evidence for the possible contribution of MPs in human diseases is still limited and this field of study in medicine is in an initial state. However, increasing studies on their toxicity in vitro and in vivo suggest worrying effects on human cells mainly mediated by oxidative stress, inflammation and fibrosis. Nephrological studies are insufficient and evidence for the presence of MPs in human kidneys is still lacking, but the little evidence present in the literature has demonstrated histological and functional alteration of kidneys in animal models and cytotoxicity through apoptosis, autophagy, oxidative stress and inflammation in kidney cells. Overall, the manuscript we report in this review recommends urgent further study to analyze potential correlations between kidney disease and MPs’ exposure in human.

## 1. Introduction

Microplastics (MPs) are defined as plastic fragments (fibers, particles or films) between 1 µm and 5 mm of size, while nanoplastics are below 1 µm. MPs can be categorized as primary or secondary. Primary MPs are small plastic pieces produced as such, while “secondary MPs” are plastic fragments deriving from the corruption and fragmentation of larger plastic pieces through physical, chemical or biological processes [1]. MPs occur in the environment through many routes: abrasion and erosion, wear and tear pf polymers, agricultural practices, industrial waste, fishing activities, garbage patches and the textile and clothing industry. Textile microfibers are one of the most important sources within primary microplastics. Laundering 6 kg of synthetic materials could release between 137–728 fibres per wash [2]. Microfibers can enter the environment throughout the supply chain of the fashion industry, from textile fabrication and clothing manufacturing to including domestic laundry activities [3]. Plastic production every year is increasing, and reached 368 million tons/year production worldwide in 2020 [4].

MPs have been detected in soil, fresh water, sea water and the atmosphere, and they have also been found in seafood, sea salt and various foods [5]. They have definitively entered into the human food cycle, becoming ubiquitous and representing an environmental threat [6]. A study performed by Cox et al. estimated an annual ingestion of 39,000–52,000 particles and a global exposition of 74,000–121,000 per year [7]. Moreover, the presence and detrimental effects of MPs in marine fauna have been demonstrated. Several studies performed on animals showed the ability of MPs to cross cell membranes and deposit on organs and tissues, causing oxidative stress, inflammation and metabolic disorders [8].

In consideration of this increasing evidence, MPs have become a growing medical concern in recent years and increasing studies have been published investigating the presence of MPs in human tissues and fluid. Surprisingly, almost the entirety of the studies showed the presence of MPs in organs and tissues analyzed such as the placenta, lungs, liver and blood, but their potential health-damaging consequences are still being debated. Moreover, an increasing amount of research on the toxicity of MPs on human cells and animal models has been carried out, and several studies into human cell lines have shown inhibition of cell growth and various molecular alterations [9]. Processes through MPs reaching organs and tissues are still a matter of research: nanoparticles can cross cellular membranes, while for bigger particles, mechanisms of endo and exocytosis are probably more involved. A pivotal role seems to be played by the formation of a coating of eco-coronas composed of biomolecules (e.g., proteins, lipids) that enhances the internalization of MPs into cells [10].

In nephrology, interest in the possible consequences of MPs’ exposure is growing too, but evidence is still scarce. Our research group is performing various research investigating the presence of MPs in the urine and kidneys and exploring their toxicity in human kidney cells also by the support of omics techniques (data still not published). To date, there is only one study investigating the presence of MPs in human kidneys, but it has not shown significant results.

In this review, we provide an overview of the recent findings of the presence and deposition of MPs in human organs, tissues and fluid. Moreover, we report the most interesting findings on the toxicity of kidneys due to MPs in vitro or in animal studies. The aim of this review is to arouse the interest of the nephrological community on the presence of MPs in humans, the consequences of these nephrological concerns and the possible implications on kidney health.

## 2. Evidence of Microplastics in Human 

In 2021, Ragusa et al. [11] and Braun et al. [12] provided the first evidence for the presence of MPs in the human placenta. Thereafter, a “challenge” emerged to find and characterize MPs’ burdens in different human organs, tissues and fluids. Various studies have been performed on the placenta [13,14,15], lungs [16,17], liver [18], blood [19], heart [20] and human body fluids [21]. 

Different methods have been employed to perform studies on MPs in humans (Figure 1). Firstly, plastic-free protocols have been adopted and improved for handling samples to attenuate contaminations (e.g., glass jar and containers and handling samples under laminar flow hoods). Different pre-treatments of samples have been developed to digest samples (mostly based on KOH and H_2_O_2_) and thus to allow analysis of microparticles [22]. Procedural blanks used as negative controls to monitor the eventual contamination of samples by MPs to avoid equivocal results and biased conclusions took on a great importance too [23]. Nevertheless, the aforementioned procedures are not usually strictly adopted altogether, thus results must be taken with caution. Different techniques are used to characterize MPs in vivo: µRaman spectroscopy, Fourier transform infrared spectroscopy (µFTIR), pyrolysis gas chromatography–mass spectrometry (Py-GC-MS). So far, the most used methods have been Raman spectroscopy and FTIR, both vibrational spectroscopy techniques based on the scattering of light and infrared waves and analysis of the resulting spectra. Recently, laser direct infrared spectroscopy (LD-IR) has been emerging as a promising and accurate method that can simultaneously analyze the whole filter and characterize particle by particle with less analytic bias [22]. MPs are ubiquitous and we are exposed daily to intake mainly by oral ingestion and airways. Indeed, various studies have detected MPs in stool, sputum and broncoalveolar lavage fluid too [24]. However, more worrying is the fact that MPs have been detected in organs, tissues, and internal fluids, corroborating the evidence for the bioavailability and deposition of MPs in the human body. Based on these recent findings, correlations with diseases have been investigated, and on the other hand, the biochemical alterations and toxicity in in vivo or in vitro studies. Shown below, we review the main data recently published on MPs’ deposition in human body and the recent evidence for the possible association with diseases (Table 1).

### 2.1. Neonatal and Gynecological System

Ragusa [11] and Braun [12] first demonstrated the presence of MPs in human placenta. In particular, the first study represents a cornerstone in MPs’ research for different reasons. They analyzed placentas differentiating between the fetal and maternal side. Moreover, they developed efficient protocols for digestion and the plastic-free handling of tissue samples that have been implemented by further studies. Based on previous studies, they employed a 10% NaOH digestion protocol for placenta samples to dissolve organic material allowing the isolation of microparticles and performed several procedural blanks for quality control. Finally, they found a moderate number of well-characterized MPs and described them using µRaman spectroscopy. This technique has been widely validated in MPs’ research and has been adopted in many studies on MPs in humans, mainly for its ability in the analysis of small particles. 

Nowadays, MPs have been studied with different methods, as described in five literature papers. In 2021, Ragusa and Braun investigated placenta with µRaman and FTIR. In both studies, polypropylene (PP) microparticles were found. By using µRaman spectroscopy, Ragusa et al. were able to characterize 12 microparticles in the range of 5–10 µm in four out of six placentas. In addition to four particles of PP, they identified different industrial pigments such as iron hydroxide oxide, copper phthalocyanine and ultramarine blue. Differently, Braun et al. analyzed with FTIR particles bigger than 50 µm. Two out of three placentas were positive for different MPs: polyethylene (PE), PP and polyurethane (PU).

More recently, three other studies have been performed on placenta using different techniques. Ragusa et al. investigated 10 human placentas using scanning electron microscopy and transmission electron microscopy [14]. MPs of 2.1–18.5 µm in size were found in all the samples analyzed. They were able to localize MPs in the villous surface intra/extracellular compartments of different placenta cellular layers and, in particular, in the lysosome, peroxisomes, lipid droplets and multivesicular bodies (intracellular), stroma, endothelial cells and pericytes (extracellular). This evidence has enhanced our knowledge on the mechanism of MPs’ intake by placenta tissues. Indeed, MPs’ presence on the villi surface corroborates the hypothesis of the transport and deposition of microparticles through maternal blood. Moreover, they found intracytoplasmic ultrastructural alterations that we will discuss more deeply below. Differently, Zhu et al. and Liu et al. investigated MPs in human placenta using laser direct infrared spectroscopy (LD-IR) [15]. They found MPs in all the 17 and 18 human placentas collected. The dimension of MPs found was similar among the studies, 80.29% smaller than 100 µm in one and in the range of 20–50 µm in 70.46% of the other. Conversely, types of MPs were different between the studies: Zue et al. preferentially found polyvinyl chloride (PVC), PP and polybutylene succinate (PBS), while Liu et al. found mainly polyamide (PA) and PU accounted for greater than 78%.

Meconium has been investigated in three different studies. Braun et al. [12] and Li et al. [25], using respectively FTIR and µFTIR, found some very different results. In the first study, two positive samples for PE and PP were found. On the contrary, using a specific treatment of H_2_O_2_ and HNO_3_, Li et al. did not find MPs on 16 meconium samples. Finally, a recent study by Liu et al. analyzed 18 meconium samples with LD-IR all resulting in being positive for MPs, mainly PA and PU. This study also aimed to identify the exposure route through questionnaires on dietary habits and plastic use. 

Ragusa et al. also investigated the presence of MPs in human breastmilk [13]. They found MPs with a size ranging from 2 to 12 µm in 26 out 34 samples using µRaman spectroscopy. Microparticles were mainly composed of PE, PVC, PP and nitrocellulose. They also investigated patients’ data (age, use of personal care product) and food habits but they did not find significant correlation with MPs. 

Collectively, these findings corroborate the hypothesis of high fetal and neonatal exposition to MPs. 

### 2.2. Respiratory and Gastrointestinal Systems

The respiratory and gastrointestinal systems represent the main entrance route for MPs. Thus, several investigations have been carried out on the exposition to MPs in these fields.

Inhalation as the intake route for airborne microparticles is corroborated by investigations in occupational illness in textile and vinyl chloride industries [33] In 2021, Amato-Laurenço et al. demonstrated the presence of MPs in 13 out of 20 human lung samples [17]. All the samples were collected by distal and proximal regions of the left lung from non-smoking adults who underwent a routine coroner autopsy. Most polymers determined with µRaman spectroscopy were PE and PP and all the particles were smaller than 5 µm, while fibers ranged between 8.1 and 16.8 µm. Differently, in 2022, Jenner et al. evidenced 39 MPs in 11 out of 13 lung tissues of smoking patients who underwent surgical procedures for cancer or a lung reduction using µFTIR [16]. Specimens were collected from peripheral regions of upper, middle or lower lobes with taking care of avoiding tumor margins. Higher concentrations of MPs were found in the lower regions and the most abundant polymers found were PP, polyethylene terephthalate (PET) and resin. MPs identified in this study were much bigger than the other with a range of particle lengths of 12–2475 µm (mean 223.1 ± 436.2 µm) and a range of particle widths of 4–88 µm (mean 22.1 ± 20.3 µm). 

Another well-conducted study was performed by Chen et al. in 100 human lung tissue samples that underwent a surgical procedure for pulmonary cancer [32]. They investigated MPs using µFTIR, LD-IR and µRaman spectroscopy. They found 65 microfibers, including 24 MPs, in 46% of the normal tissues studied. The most common types of fiber found were polyester, cotton, rayon and denim. They also found PET, acrylic and phenoxy resin in tumor tissues.

In consideration of the above-mentioned evidence and the human burden of MPs through inhalation, several studies have been performed to investigate their presence in sputum and bronchoalveolar lavage. In particular, two studies in 2023 confirmed the presence of MPs in lower respiratory tracts trough bronchoalveolar lavage analysis [24,34].

The other main way of exposure to MPs in humans is through ingestion. Indeed, the presence of MPs in table salt and drinking water is well known, as essentially their presence is in the human food chain. Thus, various studies have been performed on MPs’ presence in stool. However, to understand if MPs remain and deposit in the human digestive tract, Ibrahim et al. investigated in 2020 the presence of MPs in colectomy samples obtained by patients undergoing surgery for colorectal cancer [26]. All the 11 non-tumor colon samples tested positive for MPs with an average of 28.1 ± 15.4 particles/g of tissue. Most of the MPs characterized by µFTIR resulted in being composed by polycarbonate (PC), polyamide (PA) and PP. Further studies are needed to understand the residence time of MPs and their potential toxicity in the colon, including malignancy. 

### 2.3. Cardiovascular System

An interesting study performed by Yang et al. has recently demonstrated the presence of MPs in the human heart and surrounding tissues using LD-IR [20]. Most of the MPs’ diameters were <50 µm and 90% of the polymers were represented by PET an PU. This study is of peculiar interest because they analyzed 15 cardiac surgery patients’ different parts of the cardiac system separately: pericardia, epicardial adipose tissues, pericardial adipose tissue, myocardia and left atria appendages. Moreover, they analyzed blood sample before and after the surgery. MPs were not found in all the samples, but they have been found in all the five types of cardiac tissues. Tissue samples contain a significantly higher amount of MPs than the corresponding blood samples, corroborating the hypothesis about the potential bioaccumulation of MPs. They showed that the type and size of MPs in human blood is different compared to tissues and markedly changes in the blood before and after surgery. The most prevalent polymer in pre-surgery blood was PET, while it was PA in after-surgery blood. Finally, in post-surgery MPs, the prevalent diameter was smaller (20 to 30 µm vs. 30 to 50 µm).

Other studies investigated the deposition of microparticles in the human cardiovascular system. Wu et al. investigated the presence of MPs in human arterial thrombi using µRaman spectroscopy [27]. In their study, 16 out of the 26 samples analyzed resulted in being positive for MPs. A total of 87 particles with a size range between 2.1 and 26 µm were found globally. Interestingly, most of the particles were characterized as pigments, in particular, 21 particles were characterized as made by phthalocyanine (predominantly copper phthalocyanine). Phthalocyanine is used as a chemical pigment in the plastic, dye and ink industry, and was also detected in placenta specimens. Other particles were characterized as low-density polyethylene (LDPE), iron compounds and metallic oxides. The researchers found a positive association between MPs and blood platelet levels. Nevertheless, whether MPs contribute or not to thrombosis has to be explored in further research.

The presence of MPs has been demonstrated also in human vein tissue. Rotchell et al. detected MPs in four out five vein samples using µFTIR with a mean concentration (adjusted for contaminants) of 14.9 ± 17.2 MP/g of tissue [28]. MPs were preferably fragments of a mean length of 120 ± 227 µm (range 16–1074 µm) and a mean width of 41.3 ± 62.8 µm (range 7–300 µm). The most abundant polymers found were alkyd resin, poly vinyl propionate/acetate (PVAc) and a tie layer consisting of nylon EVA or ethylene vinyl alcohol (EVOH)-EVA.

### 2.4. Body Fluids 

In 2022, Leslie et al. and Pironti et al. detected MPs in human blood and urine, respectively [19,21], corroborating the hypothesis of mechanisms of absorption, bioavailability and clearance of MPs in the human body. In the first study, researchers found plastic particles ≥ 700 nm in the human blood of 17 out of 22 healthy donors using double shot pyrolysis–gas chromatography/mass spectrometry. This technique allowed the analysis of five target polymers: methyl methacrylate (PMMA), PP, PS, PE and PET. The most widely encountered was PET (50% of donors) followed by PS (36%), PE (23%) and PMMA (5%). This study did not imply MPs count per sample but their concentration that resulted in 1.6 µg/mL globally (maximum concentration PE 7.1 (µg/mL). Thereafter, Pironti et al. documented seven MPs in four out six urine samples from six healthy volunteers using Raman spectroscopy. MPs presented a size range of 4–15 µm and were found to be mainly composed of PP (4), but also PVC, PE and polyethylene vinyl acetate (PVA). This study first suggested the hypothesis of a mechanism clearance of MPs from the human body. µRaman spectroscopy has also been used by Montano et al. to investigate MPs in human semen [29]. Research found 16 MPs ranging between 2 and 6 µm in 6 out 10 healthy volunteers. The main polymers represented were PS, PP, PE and PET, but also PVC, PC and polyoxymethylene (POM) were found. 

Astounding results have been reported by Guan et al. in a recent study where 13 kinds of enclosed body fluids were analyzed: whole blood, cerebrospinal fluid and two main pathological body fluids (effusions and cyst fluids) [30]. The enclosed body fluids analyzed covered eight body systems and were investigated using µRaman spectroscopy in 104 patients. An extraordinary number (702) of microparticles was found despite the theorical protection given by biological barriers and membranes. The microparticle size range was between 2.15 and 103.27 µm. Microparticles were categorized in five different groups: synthetic, iron compounds, iron-free minerals, carbon and undocumented. Twenty-three MPs ranging from 19.6 µm to 103.3 µm, including nine types of polymers, were detected. PS and PP resulted in being the most represented MPs followed by PVB, PA and LDPE. The evidence of a considerable amount of MPs in cerebrospinal fluid suggests a limited barrier effect of the blood–brain barrier and the consequent deposition in terminal tissues and basically their ubiquity in human organs and tissues too. 

### 2.5. Correlation between MPs and Diseases 

The detection in human samples and healthy volunteers has stimulated research in correlating the presence of MPs with diseases or in specific clinical settings. The above-mentioned study by Chen et al. explored the correlation between MPs in human lungs and ground glass nodules (GGNs) [32]. Pulmonary GGNs are areas of lesions of homogeneous density with a hazy increase in density in the lung field as identified on computed tomography. Their detection is increasing thanks to the improvement of diagnostic methods but researchers suggest a contribution also of genetic and environmental factors. Pulmonary GGNs are observed in pre-invasive lesions or in malignancies. Using LD-IR, Chen et al. compared the frequency of detected microfibers in the GGNs and adjacent normal lung tissues. In 50 pairs of sample microfibers, 29 were found in tumor and 23 were found in normal tissues. Moreover, a total of 38 microfibers were detected in tumor tissues (58.46%) against 27 microfibers in normal tissues (41.54%). Collectively, the detection rate was significantly higher in tumors. Additionally, the type of polymers was different among tissues, with acrylic, PET and phenoxy resin present only in tumor tissues. 

Furthermore, an interesting study published in 2022 found the presence of MPs in the liver of patients with cirrhosis but not on those without underlying liver disease. Cirrhosis is one of the most relevant causes of death worldwide, regardless of its etiology (HBV/HCV, alcoholic liver disease, non-alcoholic fatty liver disease). Horvatits et al. analyzed six patients with cirrhosis and five individuals without liver disease [18]. A total of 17 samples (11 liver, 3 kidney and 3 spleen samples) were analyzed using fluorescent microscopy and Raman spectroscopy. Samples from non-cirrhotic individuals tested negative for MPs while in cirrhotic liver tissues, six different polymers with a range size between 4 and 30 µm were found. Most commonly, the polymers found were PS, PVC and PET, but also PMMA, POM and PP. These results showed a significant relationship between cirrhosis and MPs’ accumulation in the liver. Researchers hypothesized that liver cirrhosis could be a risk factor for MPs’ accumulation in the human liver, but their role in the pathogenesis of fibrosis has to be evaluated with further studies. Moreover, it is likely that the small sample sizes of this study did not allow the detection of MPs in other tissues. Nonetheless, based on these findings, we can presume that MPs will be lower in non-cirrhotic livers. 

Another interesting study investigated the correlation between MPs’ exposure and inflammatory bowel disease (IBD) in more than 100 individuals [35]. Yan et al. found that the concentration of MPs in the feces (dry mass) of IBD patients was significantly higher compared to healthy people (41.8 and 28.0 items/g dm). MPs were characterized using µRaman spectroscopy. The amount of MPs ≤ than 50 µm were in IBD were significantly higher in IBD patients compared to healthy subjects. Furthermore, the relative concentrations of the polymers identified were different between the two groups, in particular, the relative abundance of PET resulted significantly higher in the IBD group. The researchers also investigated the relationship between IBD status and fecal MP concentration through the Harvey–Bradshaw index and Mayo score, for Crohn’s disease and ulcerative colitis, respectively. A positive correlation was found with a severity of IBD in both the diseases. These data suggest that MPs could be associated with the occurrence or development of IBD, otherwise the disease and its severity have an effect on the retention of MPs through the gastrointestinal system.

Finally, MPs have shown effects on the genera of placenta and meconium microbiota [31]. Microbial communities are among the first living things to interact with MPs. Functional changes in aquatic microbiomes can alter carbon metabolism and food webs, with unknown consequences on higher organisms or human microbiomes and hence health [36]. In an infant gut in vitro model, MPs’ exposure alters the composition of gut microbiota by increasing potentially pathogenic populations of *Dethiosulfovibrionaceae*, *Enterobacteriaceae, Oscillospiraceae* and *Moraxellaceae* while decreasing *Monoglobaceae* [37]. Liu et al. first evidenced that placentas and meconium were dominated by different genera [31]. Moreover, they found various MP types and showed that PS-MPs were significantly correlated to changes in meconium microbiota, while PE correlated with the modification of placenta microbiota and total MPs, PA and PU with several genera of meconium microbiota. In particular, they found that PE was inversely correlated with *Bacteroides,* suggesting a possible antibacterial effect. Another interesting correlation was found between meconium microbiota and the particle size of MPs (50–100 µm). PA and PU are widely used in textile products which can release microparticles during washing and use and lead to indoor exposure of MPs. This study was the first to show alterations of microbiota genera in humans. Moreover, they found high concentrations of MPs in meconium, suggesting various intrauterine exposure sources beyond the transfer of MPs from the placenta to the embryo or fetus.

Nonetheless, evidence for the possible contribution of MPs to diseases is still lacking and further and urgent studies are needed to investigate their contribution to the development and maintenance of different diseases, especially new disorders, with a growing trend of incidence and new diseases with unknown etiology, with a high incidence in specific region.

Collectively, the above-illustrated studies support the presence and deposition of MPs in the human body (Figure 2). Regardless, there are marked differences among the results on the type of polymers found, the morphology and size of microparticles, within the same tissues or organs too. Rotchell at al. suggest that the distribution of different types of MPs may be tissue specific [28]. In our opinion, the above-reported evidence does not support any reasonable hypothesis on the explanations behind the differences among the studies on type of polymers, morphology and size of MPs found in different organs and tissues. There are many confounders and limitations of the studies published until now and any hypothesis will result in being very speculative. Firstly, studies are still too few and frequently characterized by very small populations studied and sample size. Moreover, almost every study used different protocols for sample collections, handling and digestions/pre-treatment of specimens. Furthermore, different techniques are used (µRaman spectroscopy, µFTIR, LD-IR, Py-GC-MS) and among the same procedure techniques of analysis are not standardized yet. Finally, differences in the presence of MPs in tissues and organs may reflect geographical characteristics and differences in MPs’ pollution in the environment. Further studies are urgently needed with larger populations studied with standardized protocols for sample treatment and improvement in analysis techniques.

## 3. Microplastics Effects on Kidney Tissues and Cells

The growing concern among the medical community about the impact of MPs on human health has influenced the increasing number of studies performed in the last few years on human tissues and cells. Even though this trend refers to nephrology too, robust evidence on the effect of MPs and its compounds on kidney cells and tissues is still lacking.

Most of the studies performed on nanoplastics (NPs) and MPs’ toxicity on kidney have been published in the last two years. Li et al. showed that PS-NPs worsen lipopolysaccharide-induced apoptosis in mouse kidney cells through the oxidative stress–endoplasmic reticulum pathway [38]. Furthermore, Tang et al. demonstrate nephrotoxicity in mice related to inflammation, oxidative stress and lipid disturbance [39]. After 6 weeks’ exposure to PS-NPs, the murine kidney index resulted in being decreased and presented tubular atrophy, glomerular collapse and inflammatory cells’ infiltration leading to decreased kidney function. Although some studies investigated NPs and MPs together, they have very peculiar characteristics due to their different size. Stability in biological solutions, formation of protein corona, absorption and intake by cell and tissues are markedly different among nano- and microparticles, making difficult comparisons between their effects in vivo and in vitro. Thus, we will focus in this paragraph on the evidence supporting a potential toxic effect of MPs on kidneys (Table 2). A first pioneering study performed by Deng et al. in 2017 investigated the tissue accumulation and toxicity of PS-MPs using fluorescent microspheres of 5 and 20 µm in mice [40]. They demonstrated kidney deposition of MPs in mice. Moreover, the results showed oxidative stress, energy and lipidic metabolism alteration through metabolomic analysis. PS-MPs effects were further examined in another study in kidney tubular cells (HK2) and mice. Researchers confirmed the accumulation of MPs in kidneys in mice and uptake by HK2 [41]. Results suggested PS-MPs induce an increase in mitochondrial ROS, ER stress-related proteins, inflammation and autophagy biomarkers. Meng et al. corroborated the detrimental effects of PS-MPs in kidneys of mice through oxidative stress and inflammation. Significant increases in SOD, GSH-Px and significant inhibition of CAT in mice exposed to PS-NPs and PS-MPs were observed [42]. TNF-α, IL-6, IL-10 and MCP-1 resulted in being significantly increased too. Overall, PS-MPs seemed to cause increased mortality, weight loss and histologically proven kidney damage in mice. Meng et al. confirmed inflammation induced by PS-MPs through the activation of NF-κB in another study performed on chicken [43]. They showed that PS-MPs induce the alteration of necroptosis via RIP1/RP3/MLKL signaling too. The activation of ROS by PS-MPs was also demonstrated by Goodman et al. in vitro on human embryonic kidney cells (HEK293) [44]. They used 1 µm PS-MPs at relevant environmental concentrations for a 24–72 h time frame. They showed significant morphological changes in HEK293 increased ROS expression and lowered gene expression of antioxidant markers (SOD2, CAT, GAPDH) indicating lowering glycolytic activity, thus the altered ability of HEK293 to contrast ROS. Altogether, these alterations affect the overall function of kidney cells. The cytotoxicity of PS-MPs has been detected on HEK293 within 24 h of exposure [45]. Oxidative stress and inflammation in kidney cells by PS-MPs could also be due to the inhibition of HO activity and induction of cytokines (IL-1β, IL-2, IL-6, TNF-α, TIMP-1 and 2) at low-noncytotoxic MPs concentration. At higher MPs’ concentrations (300 ng/mL), Chen et al. showed that PS-MPs induce autophagy and can diminish inflammation via NLRP-3 inhibition [39]. Furthermore, autophagy and apoptosis have been shown to be induced by PS-MPs by activating the AMPK/ULK1 pathway via ROS [46]. Indeed, ROS/AMPK/ULK1 and Ppargc1α/Mfn2 pathways resulted in being significantly increased after conditioning with PS-MPs and Di(2-ethylhexyl)phthalate (DEPH), an environmental plastic compound frequently used to strength plastic products. In real life, MPs are frequently combined with other pollutants present in the environment or are used as plasticizers and associated with polymers in plastic industrial production. As demonstrated by Sun et al., DEPH and PS-MPs have a synergistic toxic effect on kidney cells [46]. This mechanism, also known as the “trojan horse effect”, has also been investigated for organic and metallic contaminants [47]. In a mouse kidney injury model, toxicological effects of MPs combined with cadmium revealed kidney damage through oxidative stress, autophagy, apoptosis and fibrosis. In particular, 5 µm of MPs has been shown to adsorb cadmium and accumulate in the kidneys of mice inducing a severe biological response [47]: a decrease in antioxidant enzyme activities (SOD and CAT), increased autophagy markers LC3-II and autophagy early markers ATG5 and 7 and Beclin-1 and finally an increase in kidney fibrosis markers such as α-SMA, TGF-β1 and COL4A, ultimately leading to an alteration of kidney tissue structure and nephrotoxicity.

Interestingly, the chronic exposure to MPs has been investigated in a well-written study performed on mice [48]. Mice were treated with different diameters of MPs (80 nm, 0.5 μm, 5 μm). Histology analysis demonstrated that an 8-week exposition to PS-MPs induced mesangial matrix expansion, endocapillary proliferation, tubulointerstitial fibrosis and inflammatory infiltration in mice. Moreover, molecular analysis revealed oxidative stress induction through increased malonaldehyde levels and increased levels of inflammation markers IL-1β and MCP-1. Fibrosis was demonstrated histologically with Masson staining showing high-collagen deposition and by molecular investigations showing higher α-SMA expression after exposure to MPs. Moreover, serum creatinine levels resulted in the 5 μm MPs’ group compared to the control group. To explore the mechanisms behind the development of renal fibrosis in MPs’ exposure a transcriptome analysis was performed. With the contribution of Gene Ontology Analysis, chronic exposure resulted in being enriched in immune-related signals and is strongly associated with circadian rhythm [48].

## 4. Conclusions

MPs have become ubiquitous in the environment, leading to daily exposure and uptake for humans. Recent findings have suggested the ubiquitous presence of MPs in the human body too. Indeed, a growing number of studies are documenting the presence of MPs in all human tissues, organs and fluids. Moreover, several studies are highlighting the possible toxicity of MPs and their compounds on human cells and tissues through many different pathways. Their evidence on the human kidney is still not demonstrated, even if plausible. Conversely, various evidence in vitro and in vivo suggests a possible toxicity on kidney cells though oxidative stress, inflammation and fibrosis.

Based on this evidence, more efforts are needed to increase our knowledge on the possible toxicity of MPs in humans. In particular, a wide range of epidemiological studies are essential to understand a possible contribution of MPs in the developing and maintenance of kidney diseases and other tissue diseases in humans. Moreover, further studies must be directed to standardize procedures for the collection, handling and preparation of human biological samples. Uniform protocols for digestion and filtration are fundamental. Finally, comparison studies between different methods for MPs’ characterization, the extended use of blanks and a wider adoption of tools to prevent the cross contamination of samples (e.g., filter ventilation system, laminar flow hood) is of great importance for future progress in this field and to obtain further knowledge of MPs’ detection and effects in humans.

## Figures and Tables

**Figure 1 ijms-24-14391-f001:**
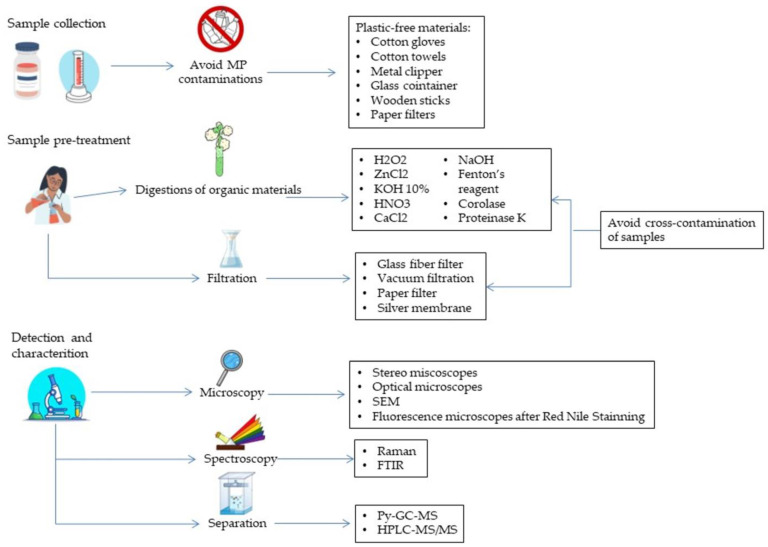
Leading steps for identification and characterization of MPs in human samples.

**Figure 2 ijms-24-14391-f002:**
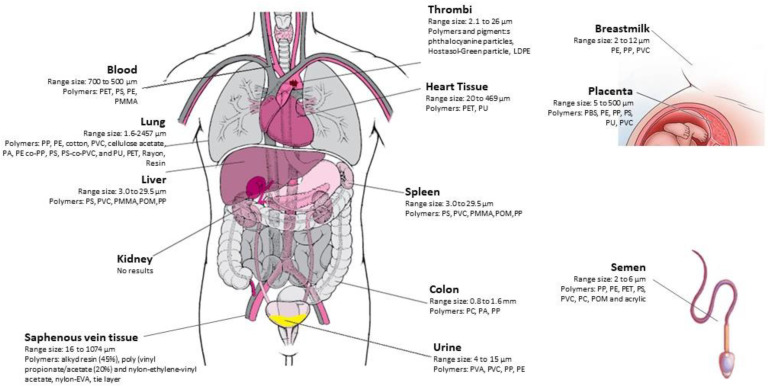
Graphical representation of main findings of MPs in human organ, tissues and fluids.

**Table 1 ijms-24-14391-t001:** Evidence for presence and deposition of MPs in human organs, tissues and biological fluids.

Author, Year	Tissue/Fluid	Method Analysis	N° Samples/ N° Patients	N°/% Particles	Particles Size Range	Polymers/Pigments
Neonatal and gynecologic systems						
Ragusa A., 2021 [11]	Placenta	µRaman spectroscopy	6 placentas, 3 samples for each placenta (from the maternal side, the fetal side and the chorioamniotic membranes)	12 MPs detected in the placentas of 4 women	From 5 to 10 μm in size	3 PP 9 pigments (Phthalocyanine, Violanthrone, Diiron trioxide etc.)
Braun T., 2021 [12]	Placenta and meconium (acquired from cesarean breech deliveries) and maternal stool post-partum	µFTIR	3 placenta samples 2 meconium 2 maternal stool post partum	5 samples resulting positive at the screening for MPs	MPs > 50 µm	More identified: PE, PP, PS, PU,
Ragusa A., 2022 [14]	Placenta	Electron microscopy with dual energy dispersive X-ray spectroscopy detectors	10 placenta samples	Presence of particles compatible with MPs inside all the different compartments of villi of all placenta samples analyzed	From 2.1 to 18.5 µm	Not specified
Ragusa, A. 2022. [13]	Breastmilk	Raman Microspectroscopy	34 breastmilk samples	58 MPs identified	From 2 to 12 µm; 47% from 4–9 µm	MPs were composed of PE 38%, PVC 21% and PP 17%.
Zhu L., 2023 [15]	Placenta	LD-IR spectroscopy	17 placenta samples	MPs were detected in all placenta samples; 11 polymer types were identified	From 20 to 307 μm, and 80.29% < 100 μm	PVC, 43%, PP, 14.5%, and PBS 10.9%.
Li Z., 2023 [25]	Meconium	Ultra-depth three-dimensional microscope and µFTIR	37 meconium samples but only 16 meconium samples were pretreated and analyzed	We did not detect MPs in any of the 16 meconium samples	/	Not identified
Pneumological and Gastrointestinal studies						
Amato-Lourenço L.F., 2021 [17]	Lung	µRaman spectroscopy	20 pulmonary tissue samples	31 synthetic polymer particles (87.5% and fibres 12.5%) 5 natural polymer particles	Synthetic polymer particles from 1.60 to 5.56 µm, natural polymer particles from 1.98 to 5.42 µm	PP (35.1%), PE (24.3%); cotton (16.2%); PVC and cellulose acetate (5.4%); PA, PE co-PP, PS, PS-co-PVC and PU (2.7%)
Jenner, L.C., 2022 [16]	Lung	μFTIR	13 lung tissue samples	39 MPs were identified within 11 of the 13-lung tissue samples	Length from 12–2475 μm, width from 4–88 μm	12 polymer types were identified: PP 23%), PET (18%) and resin (15%)
Ibrahim YS, 2020 [26]	Colon	μFTIR	Colectomy samples were obtained from 11 adults (9 from colorectal cancer 2 colon normal)	All samples had evidence of MPs with an average count of 331 per individual or 28.1 15.4 particles per g of colon tissue	Fibers from 0.8 to 1.6 mm	90% PC, 50% PA, 40% PP.
Cardiovascular studies						
Wu D., 2022 [27]	Thrombi	μRaman spectroscopy	26 thrombi	16 thrombi contained a total of 87 identified particles	2.1 to 26.0 μm	21 phthalocyanine particles 1 Hostasol-Green particle, 1 LDPE MPs; iron compounds and metallic oxides
Yang, Y., 2023 [20]	Heart tissue	LD-IR	15 cardiac surgery patients: 6 pericardia tissue samples, 6 epicardial adipose tissues, 11 pericardial adipose tissues, 3 myocardia 5 left atrial appendages, 7 pairs of pre- and postoperative venous blood samples	MPs can be detected in all five types of samples.	20 to 469 μm	9 types of MPs were identified: PET (77%) and PU (12%)
Rotchell J.M., 2023 [28]	Saphenous vein tissue	μFTIR	5 saphenous vein tissue samples	A total of 20 MP particles were identified within 4 of the 5 human saphenous vein tissue samples	Length from 16–1074 μm, and a width from 7–300 μm	5 polymer types were identified, of irregular shape (90%), with alkyd resin (45%), poly (vinyl propionate/acetate (20%) and nylon-ethylene-vinyl acetate, nylon-EVA, tie layer (20%)
Body fluid studies						
Leslie H.A., 2022 [19]	Blood	Double shot pyrolysis–gas chromatography/mass spectrometry	22 blood samples from 22 healthy volunteers	77% of donors (n = 17 out of 22) carried a quantifiable mass of plastic particles in their blood	700 and 500 nm	PET 50%, PS 36%, PE 23%, PMMA 5%
Pironti C., 2022 [21]	Urine	μRaman spectroscopy	6 urine samples of six volunteers	MPs were found in four of all the analyzed samples	4–15 µm	PVA and PVC in 1 female sample PP and PE in 3 male samples
Montano L., 2023 [29]	Semen	μRaman spectroscopy.	10 semen samples	16 pigmented microplastic fragments	2 to 6 μm	PP, PE, PET, PS, PVC, PC, POM and acrylic
Guan Q, 2023 [30]	Enclosed body fluids	μRaman spectroscope	104 patiens: 4 categories of body fluids, -whole blood, -cerebrospinal fluid - Two main pathological body fluids (effusions and cyst fluids)	Totally 23 MPs were detected in a total of 702 microparticles	19.66 to 103.27 µm	Nine kinds of MPs including: PP, PS, PTFE, PVB, PA, LDPE, PEAA, PSAN and PVA
Evidence of possible interaction with disease						
Liu S., 2022 [31]	Placenta and meconium	LD-IR	18 placentas 12 meconium samples	800 particles	From 20−500 μm (50–60% 20–50 μm)	16 types of MPs were detected in all samples. PA and PU accounted for greater than 78% of the total
Chen, Q., 2022 [32]	Lung	μFTIR	100 human lung tissues samples (including GGNs and adjacent normal tissues in patients with pulmonary GGN)	Microfibers have been found in 29 tumor and 23 normal tissues; among the detected microfibers, 24 were MPs (36.92%)	>1000 μm in length, 50% for MPs in tumor tissue, 63% for MPs in normal tissue; the widths of MPs 30–50 μm in tumor tissue, accounting for 56.25%, while <30 μm and 30–50 μm in normal tissue are both 37.5%	Cotton (which accounts for 39.47% and 51.85% of all detected microfibers in tumor and in normal tissues); rayon (constitutes 26.32% and 18.52% of tumor and normal tissues). PS (for 10.53% of tumor and 11.11% of normal tissues)
Horvatits, T., 2022 [18]	Liver (6 patients with liver cirrhosis and 5 individuals without underlying liver disease.) -Kidney -Spleen	Fluorescent microscopy and Raman spectroscopy	17 samples: 11 liver, 3 kidney and 3 spleen samples	102 MP particles; significant concentrations were detected in liver tissue samples of patients with cirrhosis	3.0 to 29.5 µm	PS, PVC, PMMA, POM and PP

Polyurethane (PU), polyamide (PA), polypropylene (PP), polystyrene (PS), polytetrafluoroethylene (PTFE), polyvinyl butyral (PVB), polyamide 6 (PA), low-density polyethylene (LDPE), polyethylene-co-acrylic acid (PEAA), polystyrene-co-acrylonitrile (PSAN), polyvinyl alcohol (PVA), polymethyl methacrylate (PMMA), polyoxymethylene (POM), polyvinyl chloride (PVC), polyvinyl butyral (PVB), polyethylene (PE), polyethylene terephthalate (PET), polycarbonate (PC) polyoxymethy (POM), polybutylene succinate (PBS).

**Table 2 ijms-24-14391-t002:** Studies in vitro and animal models on MPs effects on kidney tissues and cell.

Author/Year	Type of Study	Conditioning Type and Time	Method Analysis	Results	Conclusion
Deng Y, 2017 [40]	In vivo study: 75 five-week-old male mice	2 types of PS microspheres (diameters 5 μm and 20 μm) – Fluorescent microspheres to quantify the accumulation and distribution – Treatment groups were exposed to 5 μm and 20 μm pristine PS-MPs with the exposure doses of 0.01 mg/day, 0.1 mg/day and 0.5 mg/day by oral gavage 4 weeks of exposure	Fluorescence spectroscopy	– No increased mortality after 4 wks – The maximal concentrations of 5 μm MPs accumulated in kidney and gut were higher than that of 20 μm MPs (*p*< 0.05) – Significantly fewer 5 μm MPs were retained in liver relative to 20 μm MPs after 4 weeks of exposure (*p*< 0.05). – Inflammation and lipid droplets observed in the livers of PS-MPs-treated mice – Significant decrease in ATP level and significant increase in LDH activity in a dose dependent – Increased oxidative stress (GSH-Px and SOD increased, the activity of CAT decreased) – Significant decreases in all treatments for the levels of T-CHO and TG – Blood markers of neurotoxicity altered (AChE in liver, which decreased)	Effects on energy metabolism, lipid metabolism, oxidative stress and neurotoxic responses
Wang YL, 2021 [41]	– In vitro study: human kidney proximal tubular epithelial cell line HK-2 – In vivo study: six-week-old male C57BL/6 mice	HK-2 cells 2 lm luorescent yellow-green PS-MPs at concentrations of 0.025, 0.05, 0.1, 0.2, 0.4, or 0:8 lg = mL for 120 min or at a concentration of 0:8 lg = mL for 0, 5, 10, 30 or 60 min	– Western blot analysis – Fluorescein isothiocyanate (FITC) Annexin V/propidium iodide (PI) apoptosis detection kit – Flow cytometry	In vitro study: higher levels of mitochondrial ROS and the mitochondrial protein Bad. Higher ER stress and markers of inflammation. Cells exposed to PS-MPs had higher protein levels of LC3 and Beclin 1. PS-MPs also had changes in phosphorylation of mitogen-activated protein kinase (MAPK) and protein kinase B (AKT)/mitogen-activated protein kinase (mTOR) signaling pathways. In vivo study*:* PS-MPs accumulated and the treated mice had more histopathological lesions in the kidneys and higher levels of ER stress, inflammatory markers and autophagy-related proteins in the kidneys after PS-MPs treatment	Mitochondrial dysfunction, ER stress, inflammation and autophagy; long-term PS-MPs exposure may be a risk factor for kidney health
Meng X, 2022 [42]	In vivo study: 65 mice were weighed and randomly divided into five group	PS-NPs (50 nm) and PSMPs (300 nm, 600 nm and 4 μm) and deionized water by gavage; 24 h exposure	– Laser Scanning Confocal Microscopy (LSCM) – Transmission electron microscope – Fourier transform infrared spectroscopy (FTIR)	– Kidney weight decrease – Increased level of BUN (*p* < 0.01). – Reduction in albumin – Histological change: necrosis and detachment of renal tubular epithelium, loss of brush border, and scattered interstitial mononuclear inflammatory cell infiltrate – MPO value of the 50 nm group increased four times that of the control. – Increased SOD activity (*p* < 0.01). – Decrease in CAT activity – Increased the secretion of inflammatory factors in renal tissue (TNF-α, IL-6, IL-10 and MCP-1)	PS-NPs and PS-MPs bioaccumulated in the kidneys, and the aggregation PS-MPs exacerbated their biotoxicity; the PS-NPs and PS-MPs caused mice weight loss, increased their death rate, significantly alternated several biomarkers and resulted in histological damage of the kidney; PS-NPs- and PS-MPs- induced oxidative stress and inflammation
Zou H, 2022 [47]	In vivo study: mice	Mice were treated with Cadmium (50 mg/L) and/or 5 µm MPs (10 mg/L) for 90 days	Transmission electron microscopy – Western blotting	Tubular injury – Higher MDA levels – SOD2 and Sirt3 levels significantly elevated in the co-treatment group – Autophagy marker LC3 and the early autophagy proteins ATG5, Beclin-1 and ATG7 were significantly higher in the co-treatment group – α-SMA and COL4A1 significantly higher in the co-treatment group – e Bax/Bcl-2 ratio was increased in the co-treatment group compared with that in the Cd group, and the expression levels of their downstream regulatory proteins, Caspase-3 and Cleaved Caspase-3, were both significantly increased	MPs exacerbated Cd-induced kidney injury. MPs aggravated Cd-induced kidney injury by enhancing oxidative stress, autophagy, apoptosis and fibrosis
Goodman KE, 2022 [44]	In vitro study: Human embryonic kidney 293 cells (HEK 293)	HEK 293 treated with 5 and 100 μg/mL PS-MPs	– Flow cytometry – RT-PCR analysis	– Decline in the net metabolic activity for HEK 293 cells exposed to MPs – Significant declines in cellular proliferation rates due to microplastic exposure – Increase in ROS levels over time for each concentration – HEK showed lower expression of SOD2 and CAT for 5 and 100 μg/mL exposed cells at 24 and 72 h, decrease of GAPDH at 24 and 72 h after MP exposure	Threse morphological, metabolic, proliferative changes and cellular stress, indicate the potential undesirable effects of MPs on human health
Meng X, 2022 [43]	In vivo study: 120 chicken (1-day-old randomly assigned to 4 groups.	PS-MPs (1, 10, 100 mg/L) for six weeks, with 1 mg/L	– Transmission electron microscopy – Western blot analysis – Quantitative reverse transcription polymerase chain reaction (qRT-PCR) – RNA sequencing and bioinformatics analysis	– Mitochondrial morphology and dysbiosis (MFN1/2, OPA1, Drp1), mitochondrial structural damage by triggering imbalance in mitochondrial dynamicsAntioxidant enzyme (SOD, CAT, MDA, GSH, T-AOC) activity was significantly altered, which in turn caused oxidative stress – H&E staining results showed damage and inflammation of chicken kidney by activated NF-κB P65 and increased expression of pro-inflammatory factors (TNFα, iNOs, IL-1β and IL6) – Necroptosis through activated RIP1/RIP3/MLKL signaling pathway	This study was the first to show that oral intake of PS-MPs induced inflammation and necroptosis in chicken kidney and the differences in damage were linked to the concentration of PS-MPs
Chen YC, 2022 [45]	In vitro study: human embryonic kidney 293 (HEK29)	The HEK293 cells were treated with PSMPs (3–300 ng/mL) for 24 h.	– Western blot analysis – Histology	PSMPs can: – Adhere to the cell membrane and get entirely engulfed by HEK293 cells; – Induce cytotoxicity by oxidative stress via inhibition of the antioxidant haem oxygenase-1; – Induce depolarisation of the mitochondrial membrane potential and formation of autophagosomes confirmed that apoptosis; – Activate the inflammatory factor; – induced autophagy, which might further reduce NLRP3 expression; – Impair kidney barrier integrity and increase the probability of developing acute kidney injury through the depletion of the zonula occludens-2 proteins and α1-antitrypsin	These results demonstrated that environmental exposure to PSMPs may lead to an increased risk of renal disease
Xiong XI, 2023 [48]	In vivo study: C57Bl/6 J mice (3 weeks old; male)	H_2_O, 80 nm, 5 µm, and 0.5 µm groups according to the diameter of MPs.	– Quantitative reverse transcription polymerase chain reaction (qRT-PCR) – Transmission electron microscopy (TEM) – RNA sequencing and bioinformatics analysis	– Inflammatory response, oxidative stress and cell apoptosis in the kidney and induce kidney injury (disrupt glomerular integrity and barrier function, and cause endothelial cell damage) which ultimately promotes kidney fibrosis; transcriptome data revealed that chronic exposure to MPs could alter the expressions of multiple genes related to immune response and circadian rhythm	These data provide new evidence and potential research for investigating the harm of MPs to kidney of mammals and even humans
Sun x, 2023 [46]	In vitro study: Human kidney embryonic cells (HEK29) – In vivo study: 60 SPF-grade Kunming mice aged 6–8 weeks	4 groups: the control, DEHP, MPs, and DEHP + MPs group – DEHP group were treated with 100 μM DEHP for 24 h; – MPs group were treated with 300 μg/mL PS-MPs for 24 h – DEHP + MPs group were treated with 100 μM DEHP and 300 μg/mL PS-MPs for 24 h	– Histology – Transmission electron microscopy – Immunofluorescence – Real-time quantitative PCR, – Western blot analysis	– Significantly increased expression levels of ROS/AMPK/ULK1 and Ppargc1α/Mfn2 signaling pathway-related genes; – Upregulation of the mRNA and protein expression levels of autophagy markers	The combined exposure to DEHP and PS-MPs caused oxidative stress and activated the AMPK/ULK1 pathway, thereby inducing renal autophagy

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
