# Peer review of "Microplastics and Kidneys: An Update on the Evidence for Deposition of Plastic Microparticles in Human Organs, Tissues and Fluids and Renal Toxicity Concern"

_ijms, 2023, doi:10.3390/ijms241814391_

Round 1
Reviewer 1 Report
Authors are requested to undertake the said suggestions.

Few sentences require rewriting. Grammatical mistakes needs to revised.
Author Response
We thanks both the reviewers for and we try a tour best to improve the manuscript based on their comments and suggestions. Hereinafter we reply point-by-point to their comments.
The review focussing on the global issue related to presence of microplastics in humans and possible their occurrence in kidney. However, to an extent the authors need to clearly define the problem statement rather than broadly stating it. The manuscript clearly tabulated the data obtained from various reference articles.
We thanks again the reviewr for his global judgment and we stressed and underlined some pivotal point to identify better the problem.
Major Comments
- Line 40: Please provide statistics to the current year rather than last century. We chenge the paper accordingly
- Table 1: Reference numbers should be added. We add the references to the tables
- Section 2.1. to 2.5: Most of the information is repetitive, already present in Table 1. Try to present or discuss the data apart from what has already been discussed in Table 1. We improved a tour best the text adding (when it was possbile) new data of interest. We performed another analysis of the papers to include other interesting results. Anyway, one of the aim of the review is to provide a the detailed summary of the first evidence of MPs in human samples, in particular type of polymers and dimensions of MPs to get a global picture of these findings.
- As literature is available abundantly, authors are requested to add more figures. We performed another figure
- Should emphasize more on microplastics and their toxicity in kidneys as title indicated. We tryied a tour best to highlight this part when it was possible.
- Tables presented are good. They neatly outlined the reference articles. Thanks
- Abstract and introduction are too general. Abstract should be rewritten and should present the problem statement clearly and should outline the insights offered by the review. We tryied to improve these parts accordingly
- Authors are requested to investigate the association of microplastics occurrence and disruption in kidney functions, as literature is available. We tryied to improved this topic in the last paragraph and conclusion. Evidence a tour knowledge are still limited to studies in vitro and animal model. We stress some results specific of macro-microscopic alterations and reduction of kidney function
- Conclusion lacks perspective and should also implicate the future research prospectus. We improve this part accordingly
Minor Comments
- Line 36: Abbreviate Microplastics in Line 35.
- Line 52: “Moreover, increasing research”.
- Line 58: Rewrite the sentence.
- Line 61: “tissues, and fluid”.
- Line 75: “H2O2”.
- Line 86: “detected in organs, tissues, and internal fluids”.
- Line 108: Rewrite, “Nowadays, MPs have been investigated with different methods in five studies”.
- Table 1: Abbreviations such as “μFTIR, LD-IR” are already stated in the text. So, use only abbreviation in the table, remove full forms.
- Line 133: “H2O2 and HNO3”.
- Abbreviations should only be used if the intended full form is present more than thrice.
- Abbreviations should be indicated upon first mention in the text.
- Manuscript requires English editing. Few sentences require rewriting and minor spell checks as well as grammatical mistakes needs to be revised.
We changed the paper accordingly to the list above
Reviewer 2 Report
Porta et al. submitted the manuscript, " Microplastics and kidney: An update on the evidence of deposition of plastics microparticles in human organs, tissues and fluids and renal toxicity concerns," which highlights the microplastic toxic effect on the cardiovascular system, liver, lungs, and kidneys.
A few points need to be addressed in the revised version.
1. Authors need to consider the textile industry waste in the introduction section, as is considered a significant source of nano plastic and microplastic origin, which is underwritten in the current form, in my opinion.
2. Textile-based materials, during their life cycle, keep producing microfilaments or microfibers that wash out mostly during their washing. These microfilaments (derived from cotton or polycotton) are not biodegradable as when these textile materials get dyed with colorants, they make a covalent bond with reactive dyes and, therefore, significantly abrogate the biodegradable characteristics of cellulose-based textile materials (cotton or polycotton), and therefore exhibit a tremendous shelf life and can reach to freshwater beds or sea beds.
3. When microfilaments reach the human body (or other organisms) through ingestion or other means, they alter the microbiome of the human gastrointestinal tract; such information is currently underestimated in the manuscript.
As for suggestions, these are some studies that can considered.
(a) Exposure to polyethylene microplastics alters immature gut microbiome in an infant in vitro gut model
(b) Collateral effects of microplastic pollution on aquatic microorganisms: An ecological perspective
(c) Microplastics induce intestinal inflammation, oxidative stress, and disorders of metabolome and microbiome in zebrafish
(d) The distinct toxicity effects between commercial and realistic polystyrene microplastics on microbiome and histopathology of the gut in zebrafish.
The paper is well-organized, representing one of the significant issues in the material science field, and must be considered for the current journal.
Author Response
We thanks the reviewer and we improved the text based on his suggestions and references. We underlined the importance of textile-based material in MPs pollution and we ehance the results available on microbioma and MPs